# A Multi-Feature Fusion Method for Urban Functional Regions Identification: A Case Study of Xi'an, China

Zhuo Wang, Jianjun Bai  and Ruitao Feng *

School of Geography and Tourism, Shaanxi Normal University, Xi'an 710119, China;
wang-zh@snnu.edu.cn (Z.W.); bjj@snnu.edu.cn (J.B.)
* Correspondence: feng-rt@snnu.edu.cn

**Abstract:** Research on the identification of urban functional regions is of great significance for the understanding of urban structure, spatial planning, resource allocation, and promoting sustainable urban development. However, achieving high-precision urban functional region recognition has always been a research challenge in this field. For this purpose, this paper proposes an urban functional region identification method called ASOE (activity–scene–object–economy), which integrates the features from multi-source data to perceive the spatial differentiation of urban human and geographic elements. First, we utilize VGG16 (Visual Geometry Group 16) to extract high-level semantic features from the remote sensing images with 1.2 m spatial resolution. Then, using scraped building footprints, we extract building object features such as area, perimeter, and structural ratios. Socioeconomic features and population activity features are extracted from Point of Interest (POI) and Weibo data, respectively. Finally, integrating the aforementioned features and using the Random Forest method for classification, the identification results of urban functional regions in the main urban area of Xi'an are obtained. After comparing with the actual land use map, our method achieves an identification accuracy of 91.74%, which is higher than other comparative methods, making it effectively identify four typical urban functional regions in the main urban area of Xi'an (e.g., residential regions, industrial regions, commercial regions, and public regions). The research indicates that the method of fusing multi-source data can fully leverage the advantages of big data, achieving high-precision identification of urban functional regions.

**Keywords:** urban functional regions; multi-source big data; social sensing; feature integration; ASOE (activity–scene–object–economy)

## 1. Introduction

The accelerated advancement of urbanization has brought new challenges to the planning and layout of urban spatial structures, highlighting the supportive value of research on the identification of urban functional regions. Urban functional regions refer to the spatial distribution of various functional activities within a city and the corresponding differentiation of neighborhoods generated by them. Specifically, they represent the areas within a city that are designated for economic or social activities such as commerce, public, industry, and residence [1]. A rational urban spatial structure is a necessary condition for achieving high-quality urban development. Accurately identifying urban functional regions and having a clear urban spatial structure are of significant importance for proper urban spatial planning and sustainable urban development [2–4].

The traditional identification of urban functional regions relies on land use planning maps and questionnaire surveys. This method requires significant human and material resources. Analyzing functional regions using images or text not only requires a large amount of data from various sources but also the information obtained from these single data sources is limited. Apart from traditional methods, early research on functional region identification often emphasizes the utilization of static spatial location data, such as Point

of Interest (POI) [5,6], location-based positioning data [7], and remote sensing imagery [8,9]. Relying solely on basic geographic information such as the classification categories of points of interest (POI) and the spatial distribution of positioning points has limitations in terms of extracting data features, and it is also susceptible to interference from surrounding noise. Leveraging remote sensing technology, traditional pixel-based and object-based classifications can capture relevant features representing the physical characteristics of urban surfaces. However, these techniques primarily focus on perceiving the natural attributes of cities, such as buildings, grasslands, lakes, and wastelands, and are unable to effectively detect urban functional regions characterized by various socio-economic attributes. Consequently, they still fall short of achieving high-precision identification of functional regions. With the development of remote sensing technology and the improved accessibility of multi-source data in the era of big data, traditional methods for identifying functional regions have been greatly supplemented to address limitations such as single data features and high data acquisition costs, thus opening up new avenues for the automatic identification of urban functional regions [10]. Among them, the fusion of social sensing data and remote sensing imagery for scene classification has become a major research focus.

Social sensing refers to individual-level geospatial big data and the associated analytical methods. This concept was first introduced in the literature [11], where the authors established a research framework based on big data to extract human spatial behavioral patterns and characterize spatial variations. According to this research framework, although individual behaviors may appear random, the massive collective behavior reflected in big data often exhibits regular patterns, which are closely associated with geographical environmental characteristics, especially socioeconomic factors. Therefore, leveraging geospatial big data enables the process of inferring information about the land "from people to the land", which partially compensates for the limitations of traditional remote sensing techniques that primarily focus on perceiving natural geographical features. Based on this, researchers have utilized social sensing data, including social media such as Weibo [12–14], travel trajectories [15–19], mobile phone signals [20–22], call detail records [23], social statistics, and on-site survey data [24], in combination with static POI data or urban landscape data [25]. This integration has led to improvements in classification accuracy in relevant studies, thereby validating the effectiveness of incorporating social sensing data. On the other hand, the purpose of scene-level classification of remote sensing images is to semantically classify each image based on its content, thereby gaining a comprehensive understanding of the overall content and underlying meanings of the imagery. This approach can make a significant contribution towards achieving better classification results [26]. With the improvement of spatial resolution and availability of remote sensing imagery, as well as the maturity of deep learning algorithms, scholars have proposed methods that integrate high-resolution remote sensing imagery to extract its advanced semantic features for functional region recognition [27–30]. These methods have achieved good recognition results. The related studies also employ classification algorithms such as XGBoost [31] and random forest [32] as classifiers and obtain ground truth data for validation purposes. This allows for supervised verification of the overall accuracy of the classification results.

In fact, integrating social perception data and high-resolution remote sensing images to identify functional regions has become quite common. Current research focuses on two main aspects: firstly, improving data feature extraction and fusion methods. To this end, scholars have proposed methods such as multimodal deep learning approaches with attention mechanisms [33], self-organizing map (SOM) neural network models based on improved dynamic time warping (Ndim-DTW) distances [34], context-coupled matrix factorization (CCMF) considering contextual relationships [35], and the adoption of Synthetic Minority Over-sampling Technique (SMOTE) to mitigate the impact of data imbalance [36]. Secondly, addressing the spatial heterogeneity of functional region units, also known as the scale effect problem. Some scholars have constructed multi-scale quantitative interpretation frameworks for functional regions based on mobile phone data, taxi trajectories, and road

network data from the perspective of human-land interaction [37]. Others have developed recursive models for different levels of urban road networks to classify multi-scale functional regions [38]. Some have proposed a hierarchical spatial unit partitioning method, dividing the research area into many hierarchical units while considering the degree of mixture in each unit. At a finer scale, these research methods and improvements mentioned above further enhance the efficiency of identifying urban functional regions [39].

From the research outlined above, the identification of urban functional regions has evolved from simple single data analysis to the integration of multiple data sources, and further to the refinement of data mining methods and the study of identifying multi-scale functional regions. However, the main challenge in urban functional region identification research lies in the lack of integrated frameworks that can deeply explore and effectively integrate multiple sources of data reflecting urban characteristics [1,24,31].

Towards this end, this paper proposes a method for integrating and processing multimodal data. It utilizes remote sensing images, building footprints, points of interest (POI), and Weibo data, which exhibit functional region differences. The method aims to fully explore the underlying semantic features contained within the multiple data sources, enabling high-precision identification of four typical functional regions in the main urban area of Xi'an. Our main contributions can be summarized as follows: Firstly, the proposed method extensively explores the scene features of remote sensing images, the object features of buildings, the socioeconomic features of POIs, and the human activity features of Weibo texts. The research perspective transitions from static to dynamic, and from human to land, thereby extracting important semantic information closely related to urban functional regions. This ultimately enables high-precision classification and identification of functional regions in the study area. Secondly, this paper quantitatively compares the weights of different data features when using single data sources versus multiple data sources, thus providing profound insights into the importance of each factor in the classification task. Lastly, compared to the SOE (scene–object–economy) method, our approach incorporates Weibo data that represents human activity features and utilizes BERT (Bidirectional Encoder Representations from Transformers) to extract its semantic features. By integrating the perception of "from humans to the environment", we capture the dynamic characteristics of the city and achieve superior classification results.

## 2. Study Area and Datasets

Our research area is Xi'an, the capital of Shaanxi Province, China. As of 2021, Xi'an covers a total area of 10,096.89 square kilometers, of which the urban area is 5145.70 square kilometers. It currently governs 11 districts and 2 counties including Xincheng District, Beilin District, Lianhu District, Weiyang District, Yanta District, Baqiao District, Yanliang District, Lintong District, Chang'an District, Gaoling District, Huyi District, Lantian County, and Zhouzhi County. The permanent population is 12.873 million, of which the urban population is 10.7813 million [40]. As the ancient capital of thirteen dynasties, Xi'an has a long history and profound cultural heritage, which is an international metropolis and a national central city explicitly built by the country. In recent years, Xi'an's urban expansion has accelerated significantly. How to rationally develop new land and plan urban layout has become an important issue that urban builders urgently need to solve. This article selects the main urban area of Xi'an as the research area, including five central districts: Xincheng, Beilin, Lianhu, Weiyang, and Yanta. The spatial location and main road network in the main urban area of Xi'an are shown in Figure 1.

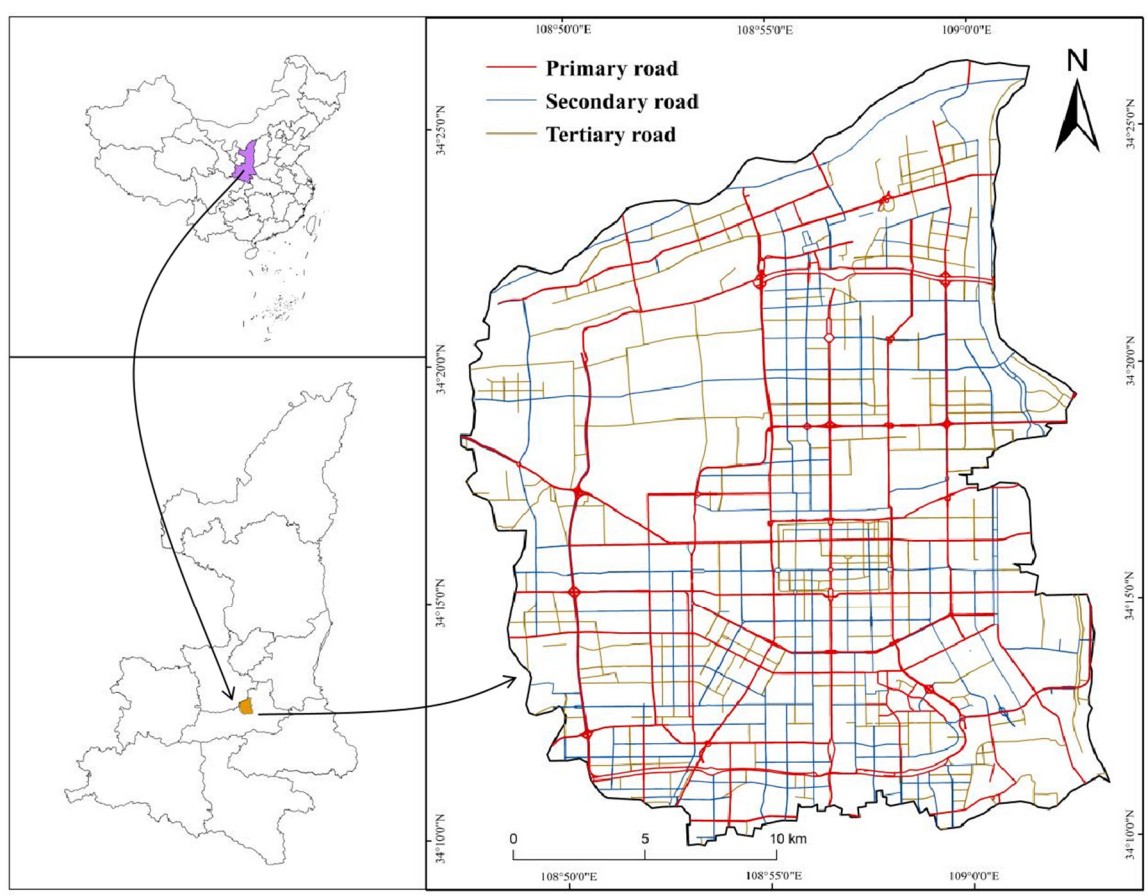

**Figure 1.** Spatial location and main road network in the main urban area of Xi'an.

Our research data mainly contains four parts: 1.2 m resolution remote sensing images, Points of Interest (POI), building footprints, and Weibo. Parcels in the EULUC-China [41] map are segmented using the OSM road network, and we use them as basic units to identify functional regions. We employ a supervised approach, relying on annotated maps from Amap and visual interpretation of images, to select parcels with a purity greater than 0.6. Here, purity refers to the proportion of the area of a specific land use. It aims to exclude samples from mixed-use regions, ensuring a more representative set of land types. Ultimately, we obtain the land use map of Xi'an, using it as the sample set and labeling values. In the end, a total of 2197 parcels are sampled: 314 for commerce, 295 for the industry, 751 for the public, and 837 for the residence. We obtained remote sensing images of Xi'an in 2020 with three spectral bands from Google Earth. The spatial resolution is 1.2 m, and the imagery covers the entire main urban area of Xi'an. The geographic coordinates range from approximately 34°6′18″ to 34°27′3″ N latitude and 108°46′34″ to 109°7′56″ E longitude. Baidu Map API is the main source of POI data for this study. We collected a total of 344,990 POI data points within the main urban area of Xi'an in November 2020, as shown in Figure 2. The kernel density map here is to better display the density distribution of POI in the main urban area, and directly displaying the original POI map is not very intuitive. The categories [42] and quantities of POI data are shown in Table 1. The POI attribute list mainly includes ID, category, GCJ02, BD09, and WGS84 latitude and longitude coordinates of the POI, administrative district, name, and street location. We utilize Python to scrape the number of floors, area, and perimeter of buildings from the AMap Open Platform (https://lbs.amap.com/, accessed on 10 August 2022), obtaining a total of 95,037 buildings in Xi'an in 2020. Weibo data are collected from the Sina Weibo Data Open Platform. We obtained a total of 20,000 Weibo data in Xi'an in August 2020. The data content includes ID, user nickname, article, sign-in point, GCJ02 and WGS84 latitude and longitude coordinates of the sign-in location, etc.

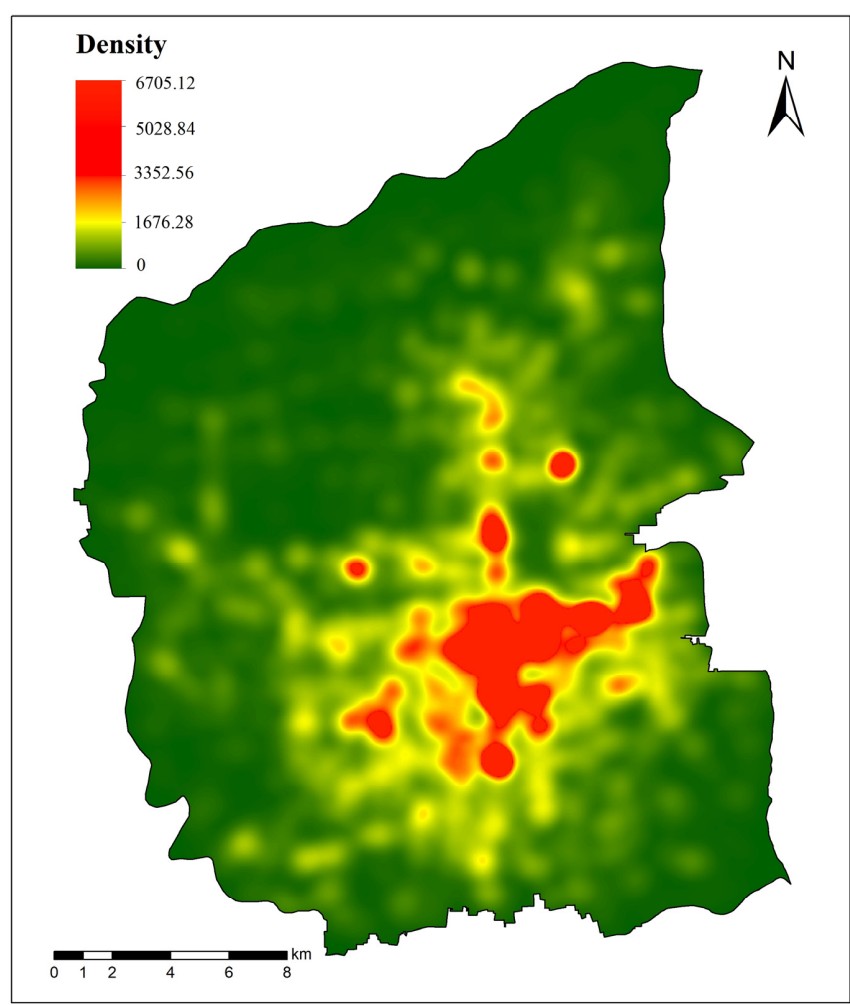

**Figure 2.** Kernel density map generated by POIs in Xi'an.

**Table 1.** The categories and quantities of POI data.

| Top-Level Category | Sub-Category | Number |
|---|---|---|
| Food | Chinese restaurant | 17,169 |
| | Foreign restaurant | 1877 |
| | Others | 11,049 |
| Education and training | Schools | 8613 |
| | Training institutions | 4286 |
| | Others | 4799 |
| Shopping | Shopping mall | 1124 |
| | Convenience store | 4466 |
| | Supermarket | 3060 |
| | Integrated markets | 10,030 |
| | Others | 51,192 |
| Companies and enterprises | Companies | 12,708 |
| | Others | 685 |
| Healthcare | Healthcare | 10,739 |
| | General hospitals | 2281 |
| | Specialized hospitals | 3034 |
| | Others | 2759 |
| Hotel | Hotels | 7008 |
| | Hostels | 2256 |
| | Others | 1980 |

**Table 1.** *Cont.*

| Top-Level Category | Sub-Category | Number |
|---|---|---|
| Real estate | Residential areas | 12,631 |
| | Office buildings | 5167 |
| | Commercial residences | 888 |
| | Others | 575 |
| Life services | Logistics company | 3707 |
| | Real estate agency | 2880 |
| | Others | 23,361 |
| Tourist attractions | Scenic spots | 2354 |
| | Park and square | 745 |
| Transportation facilities and services | Parking lots | 21,151 |
| | Bus stations | 3559 |
| | Others | 17,364 |
| Financial | Bank | 4015 |
| | ATM | 4343 |
| | Others | 1498 |
| Sports and entertainment | Leisure venues | 6211 |
| | Sports halls | 4555 |
| | Others | 2853 |
| Government institutions | Government agencies | 8912 |
| | Social organizations | 2831 |
| | Others | 4672 |
| Road ancillary facilities | Warning information | 807 |
| | Others | 83 |
| Entrance addresses | Place name | 15,514 |
| | House number | 9228 |
| Public facilities | Public restroom and phone | 6278 |
| | Emergency refuge | 196 |
| | Others | 181 |
| Motorcycle services | Motorcycle services | 153 |
| | Motorcycle maintenance | 240 |
| | Motorcycle sales | 329 |
| Car services | Car services | 12,466 |
| | Car maintenance | 2677 |
| | Car sales | 1451 |

In view of urbanization process and characteristics of Xi'an, we classify the functional regions of Xi'an into four typical categories: residential regions, commercial regions, industrial regions, and public regions in this study. Their definitions are detailed in Table 2.

**Table 2.** Definition of the four types of functional regions in this study.

| Types | Definitions |
|---|---|
| Residential regions | Living spaces for urban residents, typically comprising large-scale, densely populated residential communities that provide residential services to citizens. |
| Commercial regions | The commercial hub of the city, encompassing shopping malls, retail centers, restaurants, entertainment venues, etc. |
| Industrial regions | The center for industrial production activities in the city, typically hosting a large number of industrial enterprises or industrial parks. |
| Public regions | Public space within the city, generally including schools, hospitals, squares, scenic areas, parks, etc. |

## 3. Methodology

### 3.1. Functional Area Recognition Method Based on ASOE Features

This article contains three types of data sources: image data (remote sensing images), vector data (POI, building footprints), and text data (Weibo data). The methods for processing these modal data are different. For image data, we use the CNN network, which has increasingly matured in the field of deep learning in recent years to extract the high-level features implicit in remote sensing images. Secondly, for text data, we apply a typical text mining model to extract semantic features of Weibo text. Then, statistical methods are used to extract social–economic features related to POI and building object features related to building footprints from vector data. Finally, the previously obtained features are fused based on the parcels and input into the classifier to achieve the final classification task. The methodology flowchart is illustrated in Figure 3.

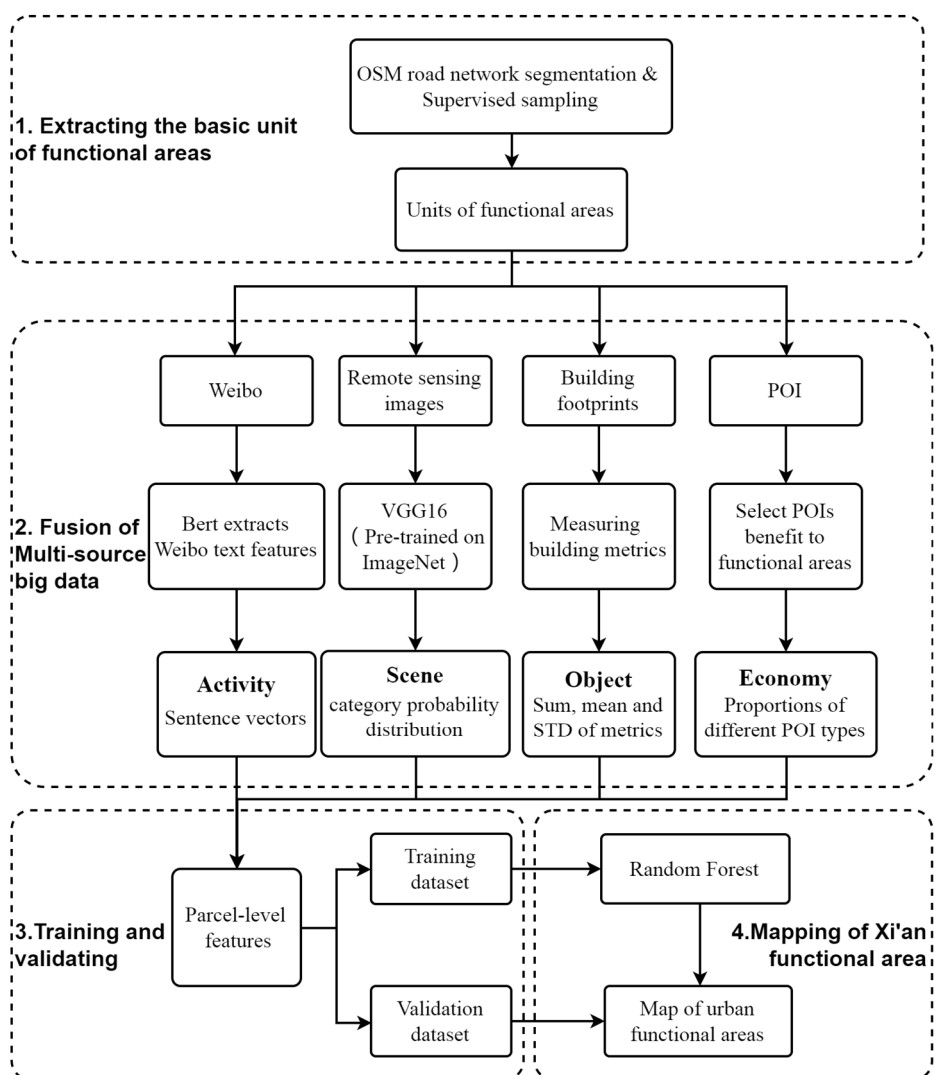

**Figure 3.** Workflow of ASOE-based methodology.

### 3.2. Extracting Image Features Based on CNN

Convolutional neural networks (CNNs) are a class of feedforward neural networks that incorporate convolutional computations and possess a deep structure. Around 2015, as CNNs were successfully applied to large-scale visual classification tasks, their applications began to emerge in the field of remote sensing image analysis [43,44]. Compared to traditional methods such as SIFT [45], HOG [46], and BoVW [47], CNNs offer the advantage of end-to-end feature learning. Additionally, they can extract advanced visual features that

handcrafted feature-based methods cannot learn. Various CNN-based scene classification methods have emerged through the utilization of different strategies with CNNs [48–50].

VGG16, as a classic CNN network, uses $3 \times 3$ convolutional kernels, which simplifies the original neural network structure [51]. This model employs a deeper network architecture, smaller convolutional kernels, and pooling sampling domains, which allows it to obtain more image features while controlling the number of parameters. This helps avoid excessive computational load and overly complex structures. The advantage of this model lies in its ability to achieve deeper layers and extract higher-level features. It consists of 16 weighted parameter layers, and its network structure is illustrated in Figure 4 [52].

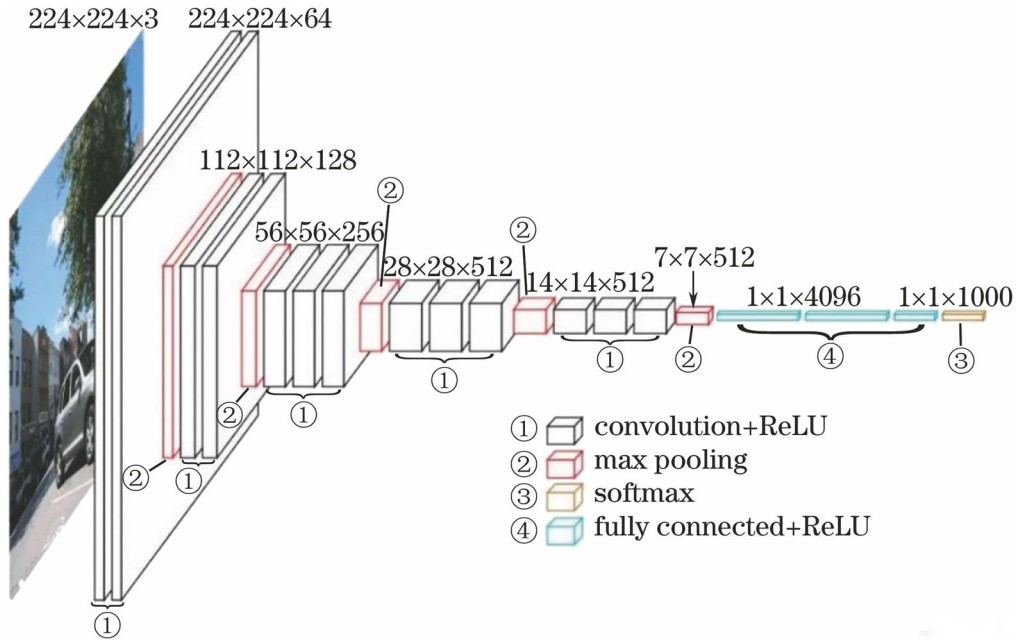

**Figure 4.** Network architecture of VGG16.

Generally, CNN-based methods for remote sensing image scene classification can be divided into three categories: using pre-trained CNN as a feature extractor [53], fine-tuning pre-trained CNN on the target dataset, and training CNN from scratch. In this paper, considering the limited sample size, we adapt the first strategy, which is to use the VGG model pre-trained on ImageNet as a feature extractor.

During the experiment, we randomly divide the remote sensing image data set at the study area scale into a training set and a validation set, with a ratio of 4:1. In the data preprocessing stage, to prevent overfitting in the training of models, we apply various forms of data augmentation methods, such as stretching, rotation, mirroring, center cropping, adjusting image opacity, hue, saturation, etc. During the network training process, in order to adapt to the classification task, we first change the penultimate fully connected layer parameters to 4. Secondly, Vgg16 pre-trained weights pre-trained on ImageNet are used to freeze all convolutional layer parameters for feature extraction, which means that the model only trains the fully connected layer, that is, only the final classifier parameters are fine-tuned. Then, we input the training data into the model and update the model hyperparameters based on the loss values of the training and verification results. The hyperparameters here include batch size, learning rate, number of iterations, etc. After that, we cross-validated the eight trained models on the validation set to obtain the optimal model with a significant classification effect. The validation performance of the optimal model is shown in Figure 5. Outside the brackets represent the labels predicted by the model, and inside the brackets are the ground truth values. The model achieves an accuracy of 88.89%. As shown in Figure 4, VGG16 has three fully connected layers. The first layer involves a nonlinear combination of locally extracted features from convolutional layers,

with a default parameter of 4096. The second layer activates the representative features from the results of the first layer using activation functions, which can express overall image characteristics. This layer has a default parameter of 1000 but is adjusted to 128 in our experiment to accommodate our dataset. The final layer uses softmax to directly output class probabilities, with a parameter set to 4. Therefore, ultimately, after cross-validation, we invoke the optimal model for training and output the feature vector of the second fully connected layer.

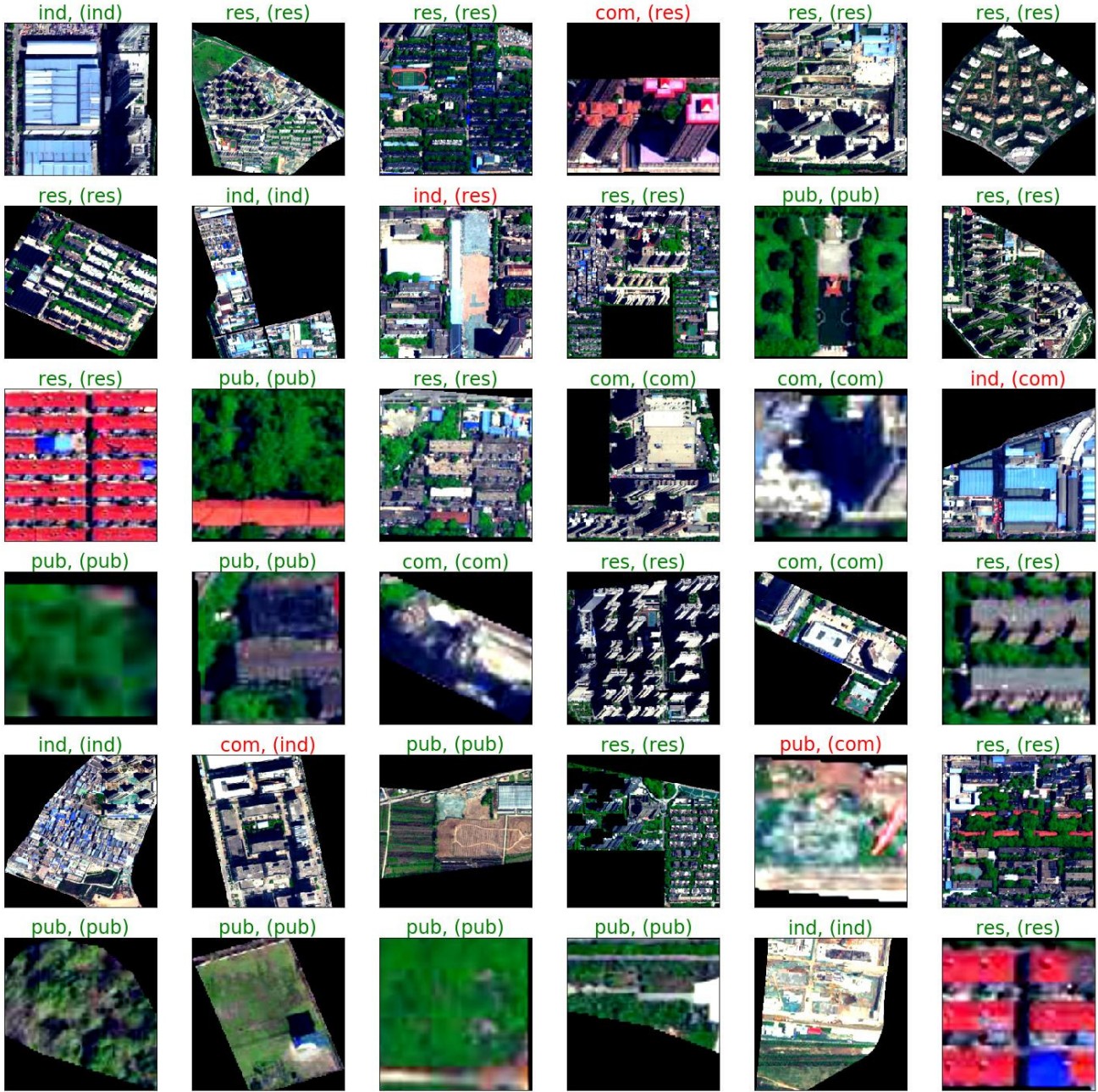

**Figure 5.** Verification effect of the optimal model (outside the brackets represents the predicted value, inside the brackets represents the true value, green letters represent correct predictions, and red letters represent incorrect predictions).

### 3.3. Abstracting Activity Features Based on Weibo Data

BERT, which stands for Bidirectional Encoder Representations from Transformers, is an overall self-encoding language model. Its objective is to utilize large-scale unlabeled

corpora for training and obtain rich semantic representations of texts, referred to as text embeddings or text representations. These representations capture the semantic information of the text. Later, these representations can be fine-tuned for specific natural language processing (NLP) tasks and applied to those tasks. The main structure of the BERT model is the transformer, as shown in Figure 6. The basic structure of a BERT pretrained model consists of the encoder part of the standard transformer, denoted as "Trm" in the figure. These encoder layers are stacked one by one to form the main body of the model [54]. The main input of the BERT model is the raw word vectors for each character/word in the text. These vectors can be randomly initialized or pre-trained using algorithms like Word2Vec to serve as initial values. The output is the vector representation of each word/phrase in the text after integrating the full-text semantic information.

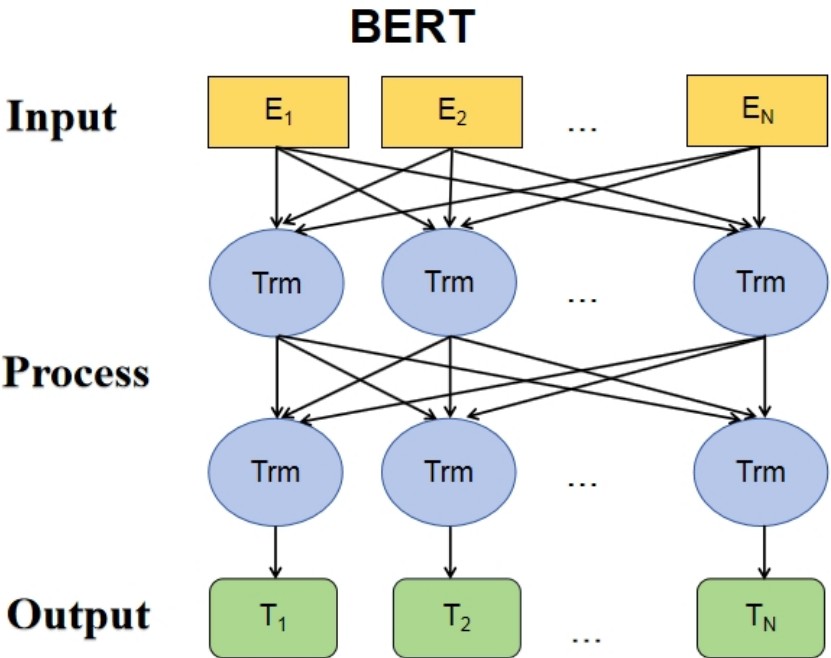

**Figure 6.** Model architecture of BERT.

Weibo data can reflect the daily activities of residents, such as shopping, travel, work, entertainment, etc. Interestingly, these activities can significantly help us distinguish the typical urban functional regions to which Weibo check-in points belong. Therefore, we utilize the BERT model to extract hidden semantic features from Weibo text data. Considering the limited amount of sample data, this paper intends to utilize the pre-trained model. This process involves vectorizing the text within each research unit, resulting in multidimensional sentence vectors that represent human activity features.

During the specific experimental process, for the 20,000 Weibo data we obtain, we first conduct data filtering and cleansing, removing Weibo data that is not within the research area or has no content. Next, we connect the land parcels with their corresponding Weibo data. Due to the sparsity of the Weibo data itself, some land parcels only contained a small amount of Weibo data. Afterward, we use the BERT model to process the Weibo text content within each land parcel. In the data preprocessing stage, it is necessary to add [CLS] and [SEP] tokens before and after each Weibo text that represents the semantics of each land parcel (e.g., [CLS] August has arrived . . . [SEP]). This is a standardization applied to the input text for the BERT model, with the purpose of treating all the text within a land parcel as a single sentence, regardless of the number of sentences within the text. During the model running phase, since the BERT model requires a threshold of 500 for the amount of text input at one time, and the number of Weibo posts in some land parcels exceeds this threshold, we conduct a second data cleaning for this part of the data. After successfully

running the model, we output and save a 50-dimensional feature vector containing the full-text semantic information of each parcel.

### 3.4. Extraction of Socioeconomic Features

Given the original POIs' diverse and overlapping categories, along with some categories having limited relevance to identifying functional regions, we conduct a reclassification, reclassifying the original 18 categories to 14. These categories include public facilities, dining services, science and education, shopping services, companies and enterprises, healthcare services, accommodation services, commercial residences, life services, scenic spots, transportation services, financial and insurance services, sports and leisure services, government agencies and social organizations. Then, using the ArcGIS10.2 spatial join method, we match these POIs with intersecting land parcels to calculate the quantity and proportion of each category, which serves as the socioeconomic characteristics of the study area.

### 3.5. Extraction of Building Features

The original building footprint data only contains information about the number of floors, building area, and perimeter. We aim to derive building metrics reflecting the differences in functional regions from these attributes in the source data. These metrics primarily include building area at an individual scale, building perimeter, building structure ratio, number of floors, and the quantity and density of buildings at a regional scale.

Area reflects the actual footprint of a building, and generally, many residential and commercial sites have larger building areas, while those on public land are relatively smaller. We calculate the total building area, average building area, and standard deviation for each region within the land parcel. Considering the standard deviation is important because in residential regions, building area differences are typically small, whereas in commercial regions, office buildings and shopping centers can vary significantly in building area. Therefore, the standard deviation faithfully captures these differences.

Building perimeter can describe the length of a building's outline and can reflect the complexity of the building's shape. Generally, residential regions have simple and relatively uniform building shapes, while commercial or public regions exhibit more complex and varied shapes with less apparent regularity. This set of metrics includes total building perimeter, average building perimeter, and standard deviation for each region.

Building structure ratio refers to the ratio of perimeter to building area. Buildings within the same region that have larger perimeters usually exhibit more complex shapes. The complexity of the shape helps distinguish between regular rectangular buildings and irregularly shaped ones. For instance, commercial and public buildings often have circular or irregular polygonal appearances, such as stadiums or shopping centers. This aids in distinguishing them from residential buildings. We measure the structural ratio for each building and calculate its sum, average, and standard deviation.

Floor is an important indicator reflecting differences in building height, which can vary significantly in different functional regions. For example, buildings intended for residential purposes in the city center typically have more floors and are consequently much taller, while industrial buildings are designed to be lower. We compile the number of floors for each building and calculate its sum, average, and standard deviation.

The above statistics are derived from an individual building perspective; however, we also recognize that at the regional scale, the number and density of buildings can reflect differences in various functional regions. For example, residential regions, being places where people live, tend to have high building densities, whereas public regions, comprising spaces like squares and parks, have much lower building densities in comparison.

As shown in Figure 7, we collect statistical data on building area, perimeter, number of floors, and structure ratio, and discuss their characteristics and patterns. Additionally, considering the building density and quantity within the parcel, we obtain a total of 14 building indicators. In the specific experimental procedure, we initially conduct a spatial

intersection between intersecting building outlines and parcels, then compute various metric values within the parcel units, and finally generate feature vectors that encapsulate building information.

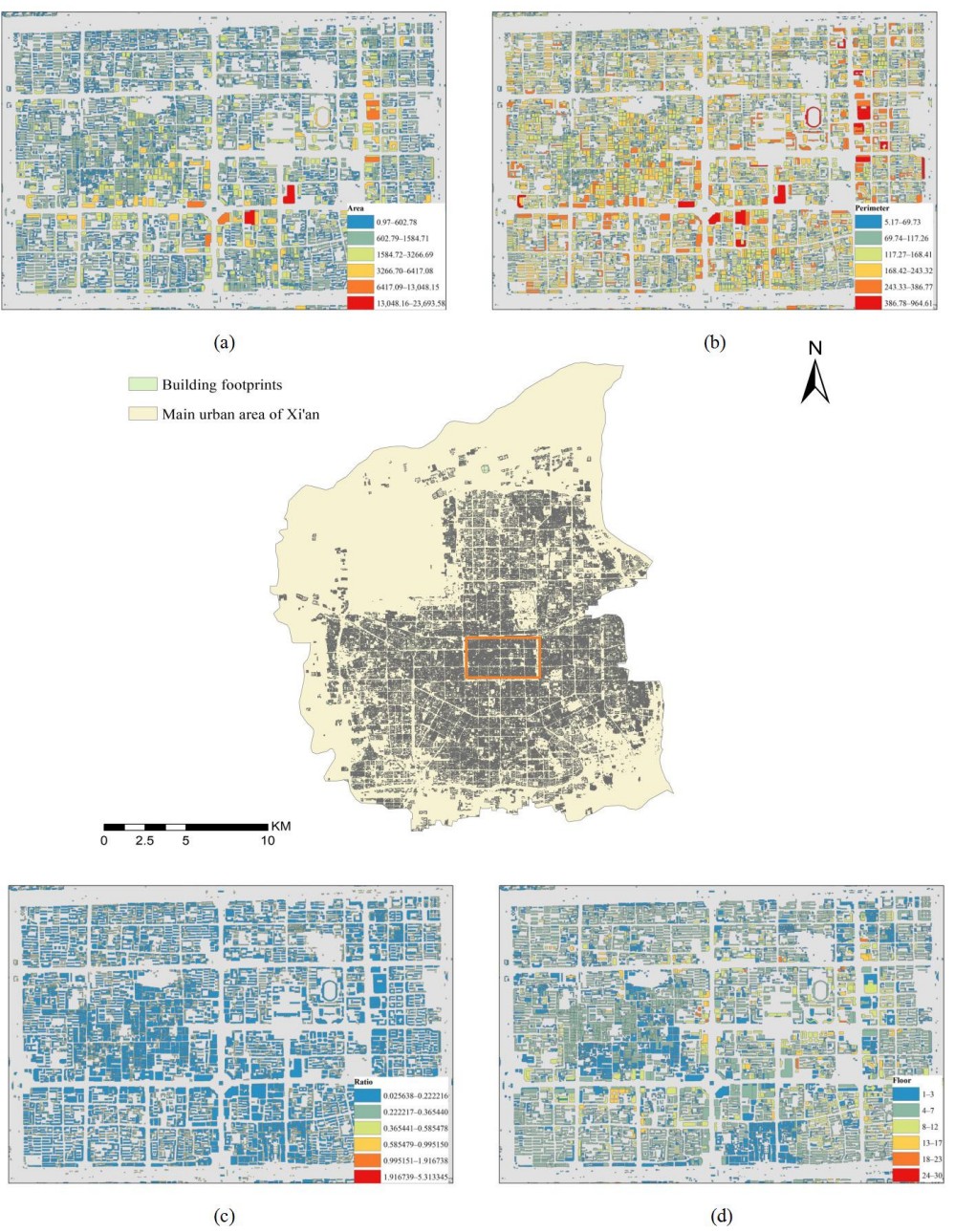

**Figure 7.** Thematic maps of building footprints. (**a**) Area of buildings. (**b**) Perimeter of buildings. (**c**) Floor of buildings. (**d**) Ratio of buildings.

### 3.6. Fusion of Multi-Source Features

At this stage, our primary task is to integrate the four types of features extracted in the previous steps to provide input for the classifier. The overall framework of this research method is illustrated in Figure 8. In the preceding feature extraction stage, we initially establish connections between land parcels and corresponding geolocated Weibo data. Subsequently, we process the Weibo text content on individual parcels using the BERT model, ultimately obtaining 50-dimensional sentence vectors that represent human activity features. By using CNN for scene detection in remote sensing images, hidden spatial information present in the images is extracted to derive scene features. The parameter

of the penultimate fully connected layer is adjusted to 128 to achieve a balance between the feature quantities in the other three data sources. For socioeconomic features, we calculate the proportions of different categories of POIs within each research unit, resulting in 14-dimensional feature vectors. By compiling data on 14 building indicators within each land parcel, we obtain the corresponding vector representing the characteristics of objects in the study area. After fusion, we obtain feature vectors with a length of 206. Finally, these feature vectors are input into a Random Forest classifier. We set the number of decision trees to 50, and the training-to-testing set ratio is set at 7:3. After the decision trees' voting process, we ultimately obtain classification results from the integration of multiple data sources.

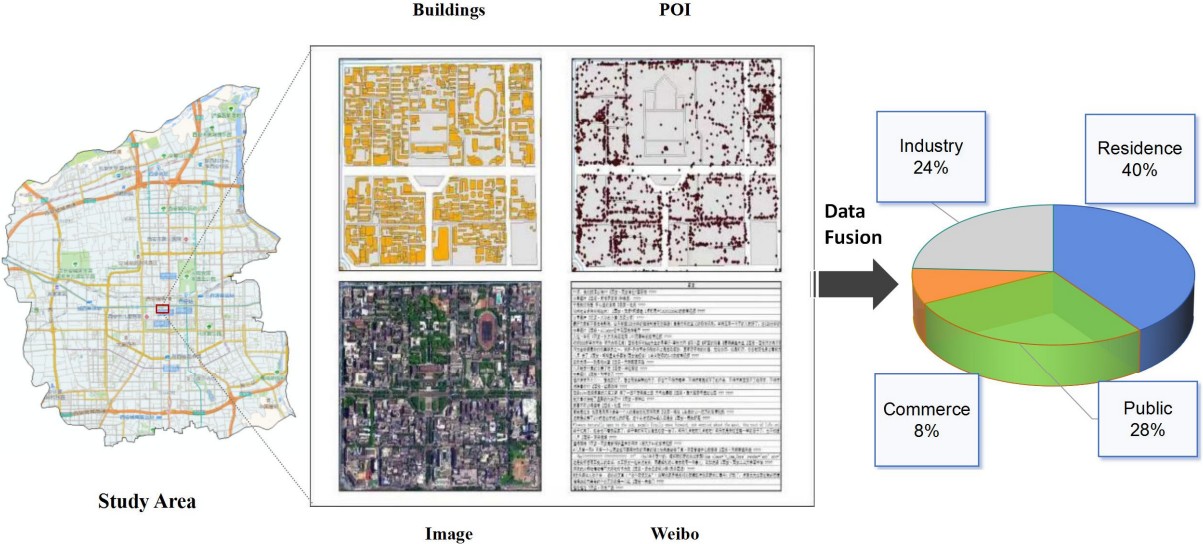

**Figure 8.** The logical structure of the ASOE method.

## 4. Results and Discussion

### 4.1. Recognition Results of Different ASOE Features

To address the issue of insufficient data feature mining in traditional functional region recognition methods, this paper comprehensively utilizes high-resolution remote sensing imagery data, POI, building footprints, and Weibo data, and employs deep learning, text mining, and statistical methods to extract features corresponding to research units and then integrates them for the final classification. Firstly, remote sensing imagery can extract advanced semantic features of regional spatial distribution based on the scene. Building footprints can provide a series of physical properties of buildings from the perspective of landscape objects. Meanwhile, open-source social remote sensing data such as POI and Weibo data can extract features related to social-economic and human activities. The identification result of urban functional regions is shown in Figure 9.

To highlight the advantages of extracting features from multiple data sources, we conduct a statistical analysis of the classification results for the functional regions obtained after merging inputs from different data sources. The comparative chart obtained is shown in Figure 10.

As can be seen from Figure 10, in single-source data classification, the best performance is the image features extracted by Vgg16, with an accuracy of 87%. Following this, POI data performs the next best, while the classification performance of buildings and Weibo data are relatively poor. In the experiments involving the fusion of multiple data sources, we observe that combining POI or building data increases the accuracy by 2.45% and 1.73%, respectively, compared to using only images as single-source input. Combining all three sources resulted in a 2.64% improvement. Furthermore, Weibo data are found to further enhance the accuracy by an additional 1.38% on top of the fusion of the first three sources. In the end, the classification accuracy achieved by fusing the four data sources reaches

91.74%, which represents a total improvement of 4.02% compared to using only images as a single-source input. From the analysis of experimental results, the disparities in accuracy among single-source data are closely linked to the quantity and effectiveness of their inherent features. For example, image features, which exhibit the best performance, have the highest quantity among the four datasets, and their expression of scene characteristics is the most effective in classification tasks. Conversely, building footprints and Weibo features have fewer quantities, and neither can effectively reflect the differences in functional areas. However, the geographic information inherent in POI data enables it to perform moderately well in reflecting functional area attributes. In multi-source data experiments, the differences in accuracy are not only associated with the quantity and effectiveness of the data's own features but also with the interaction between different data features. For instance, incorporating POI features slightly improves accuracy compared to using building footprints alone, indicating that POI data can synergize with images to produce more precise results.

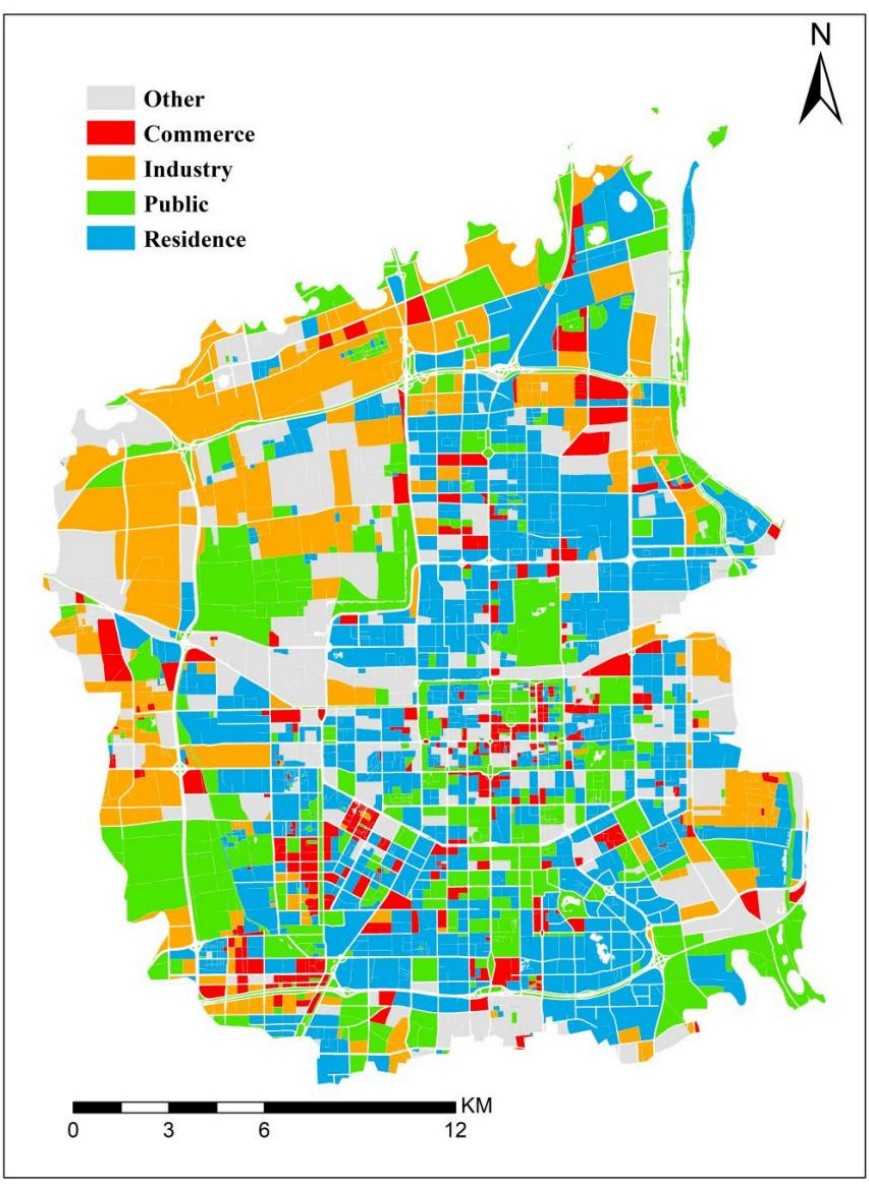

**Figure 9.** Map of Xi'an urban functional regions.

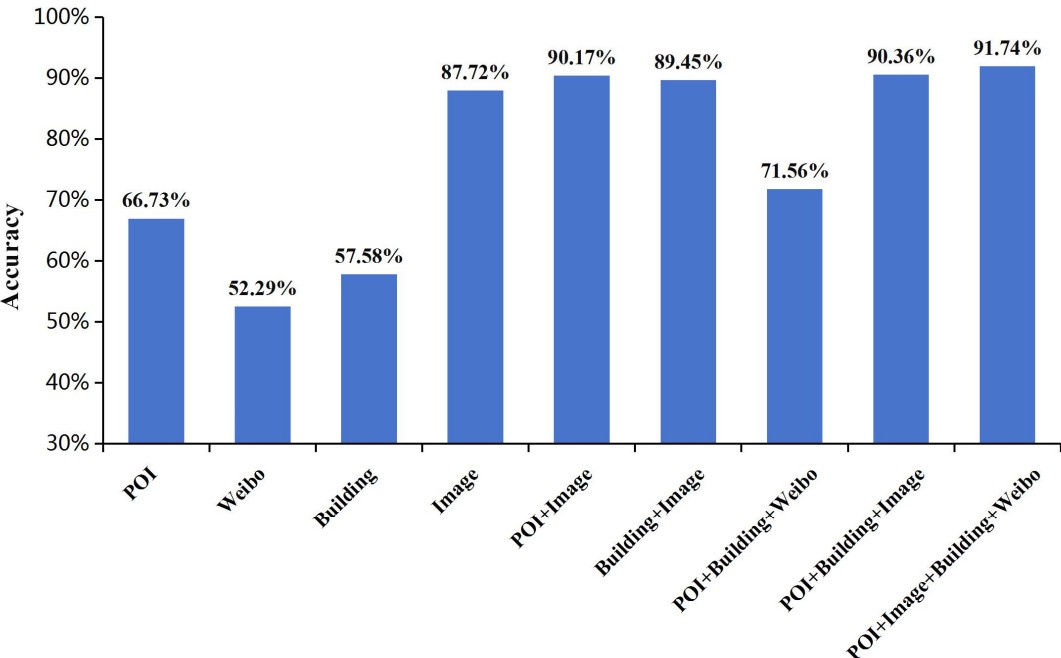

**Figure 10.** Classification accuracy of different data source inputs.

Furthermore, the classification accuracy of various functional region categories corresponding to different data inputs is shown in Table 3, and the corresponding comparative graphs are presented in Figure 11, revealing the differences in the recognition of various functional regions based on different data source features.

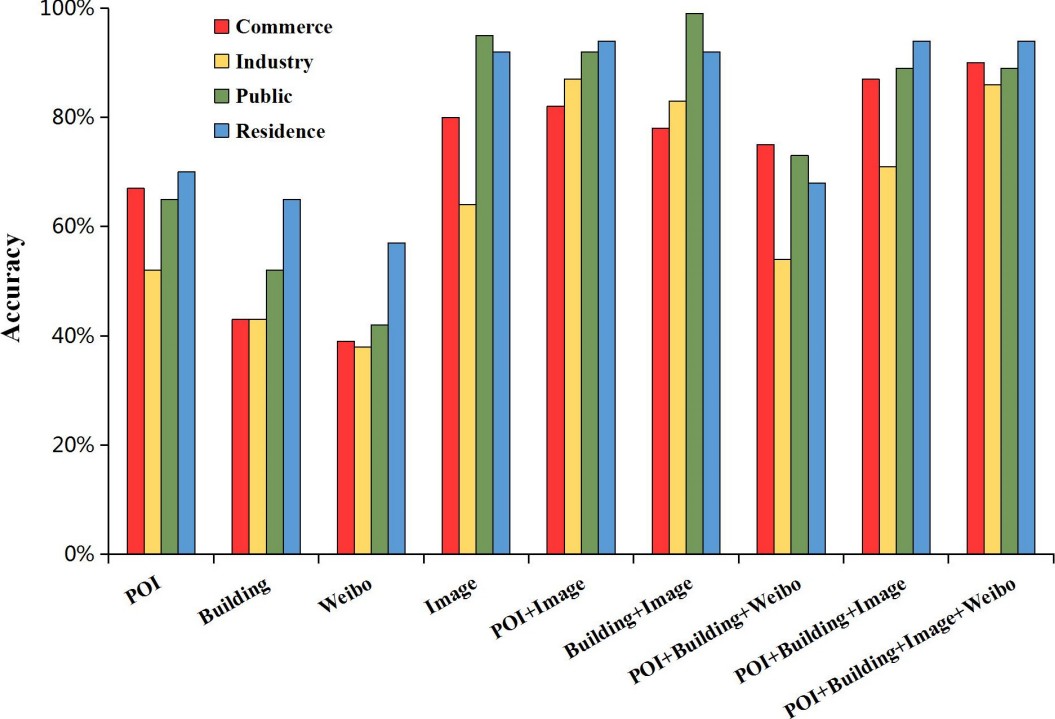

**Figure 11.** Comparison of classification accuracy for each category from different data sources.

**Table 3.** Classification accuracy of each category from different data sources.

| Input Data | Commerce | Industry | Public | Residence | Accuracy | F1 Score |
|---|---|---|---|---|---|---|
| POI | 67% | 52% | 65% | 70% | 66.73% | 0.61 |
| Building | 43% | 43% | 52% | 65% | 57.58% | 0.49 |
| Weibo | 39% | 38% | 42% | 57% | 52.29% | 0.48 |
| Image | 80% | 64% | 95% | 92% | 87.72% | 0.83 |
| POI + Image | 82% | 87% | 92% | 94% | 90.17% | 0.88 |
| Building + Image | 78% | 83% | 99% | 92% | 89.45% | 0.86 |
| POI + Building + Weibo | 75% | 54% | 73% | 68% | 71.56% | 0.68 |
| POI + Building + Image | 87% | 71% | 89% | 94% | 90.36% | 0.90 |
| POI + Building + Image + Weibo | 90% | 86% | 89% | 94% | 91.74% | 0.92 |

From Table 3 and Figure 11, in single-source data experiments, the recognition accuracy of image data is high for all regions except industrial regions, while Weibo and building footprints achieve a relatively high accuracy of 60% only in identifying residential regions. POI data performs moderately with the least difference between categories. The primary difference in multi-source data experiments lies in the identification of industrial regions. Among these, the combination of POI, building footprints, and Weibo yields the lowest accuracy at only 54%. However, combinations involving POI and image, building and image, and POI, image, building, and Weibo all perform well, with accuracies exceeding 80%. POI, image, and building achieve moderate performance with an accuracy of 71%. These phenomena may be closely related to the varying abilities of the data features to explain differences in industrial zones. Single-source data features, apart from image features, are generally weaker, but after fusion with multi-source features, they are enhanced to different extents, resulting in better performance. In particular, when compared to the classification results using only images, the ASOE method, which utilizes multi-source feature fusion, achieves an improvement of even over 10% in accuracy for identifying commercial and industrial regions, while maintaining recognition rates of around 90% in the other two functional region categories.

### 4.2. Evaluate the Contribution of Each Factor in Multi-Source Data

Random Forest is a supervised classification method that not only serves as a classifier to produce classification results but also provides the contribution of each factor within the utilized data to the classification task. When identifying using only POI, the weights for various types of points of interest are shown in Table 4.

**Table 4.** Contribution of each feature when only using POI.

| The Features of POI | Weights |
|---|---|
| (1) Accommodation | 0.122697 |
| (2) Shopping | 0.108659 |
| (3) Living | 0.103366 |
| (4) Transportation | 0.097480 |
| (5) Company | 0.086141 |
| (6) Catering | 0.078285 |
| (7) Science and Education | 0.064490 |
| (8) Medical | 0.063619 |
| (9) Government | 0.053805 |
| (10) Business | 0.053715 |
| (11) Finance | 0.053673 |
| (12) Landscape | 0.039180 |
| (13) Public | 0.038975 |
| (14) Sports | 0.035915 |

Among them, the top three features include accommodation services, shopping services, and life services. Accommodation services include hotels, motels, hostels, etc. These

points of interest are typically located near bustling commercial districts, office areas, or famous landmarks, primarily providing accommodation services to out-of-town tourists. Therefore, they contribute to identifying commercial regions and public regions related to tourist attractions. Shopping services include large supermarkets, shopping malls, specialty commercial streets, etc., which can infer that the region is likely a commercial district. However, there are also convenience stores, personal care shops, cosmetics stores, etc., open near residential regions. Life services encompass telecommunications service centers, beauty salons, job markets, logistics and delivery services, laundry facilities, etc. They are typically located within residential regions, providing services related to people's daily lives, thus making it possible to effectively identify residential regions in the city.

Only using building footprints, the weights of 14 building indicators are as shown in Table 5. The role of the standard deviation of area is the most significant, with a weight of 0.1097. Next are density and the average number of floors, with weights of 0.0942 and 0.0902, respectively. Additionally, the mean area and the standard deviation of floors also make significant contributions, with weights of 0.0862 and 0.0738, respectively. This indicates that building area and floor height contribute more information to the task of identifying and delineating functional areas. Their differences can be effectively used to distinguish the functional attributes of different regions.

**Table 5.** Contribution of each feature when only using building footprints.

| The Features of Building | Weights |
| --- | --- |
| (1) Std_Area | 0.109729 |
| (2) Density | 0.094252 |
| (3) Mean_Floor | 0.090231 |
| (4) Mean_Area | 0.086247 |
| (5) Std_Floor | 0.073887 |
| (6) Sum_Floor | 0.072016 |
| (7) Mean_Length | 0.071452 |
| (8) Sum_Area | 0.065986 |
| (9) Std_Length | 0.065733 |
| (10) Sum_Length | 0.064708 |
| (11) Mean_Ratio | 0.056086 |
| (12) Sum_Ratio | 0.053636 |
| (13) Std_Ratio | 0.048454 |
| (14) Count | 0.047583 |

Combining the two aforementioned data sources and incorporating Weibo data for classification results in an accuracy rate of 71.56%. Among these, the top ten weighted factors are shown in Table 6, consisting of six points of interest features and four building features. Transportation facilities and corporate enterprises have the highest weights in the classification task, reaching 0.045597 and 0.043930, respectively. Regarding building features, the sum of floors and perimeter contribute the most to the classification task, which shows a slight difference compared to the weight results obtained during single-source data classification. This suggests that during the process of merging multiple data sources, certain data features may interact or influence each other.

Furthermore, when incorporating remote sensing image features, there are significant changes in factor weights (due to the large dimensionality of the image features, which is 128 dimensions, it is not convenient to display directly. Therefore, here, only the summarized results are presented.). The results indicate that image features play a decisive role in classification, with a total weight of about 0.91 for image features. This suggests that image features based on DCNN provide rich and diverse urban area scene information, aligning with our expectations. When using only remote sensing images, the accuracy reaches 87.72%. However, undoubtedly, POI, building footprints, and Weibo also play a more significant auxiliary role. Using a single data source, even remote sensing images alone cannot

address the issue of uneven accuracy between categories. The addition of POI, buildings, and Weibo not only improves accuracy but also reduces the gap between categories.

**Table 6.** Top ten feature weights when merging three data sources.

| Features | Weights |
|---|---|
| (1) Transportation | 0.045597 |
| (2) Company | 0.043930 |
| (3) Shopping | 0.042038 |
| (4) Accommodation | 0.034938 |
| (5) Living | 0.034696 |
| (6) Sum_Floor | 0.034240 |
| (7) Sum_Length | 0.029728 |
| (8) Mean_Area | 0.029229 |
| (9) Std_Area | 0.027634 |
| (10) Catering | 0.024588 |

### 4.3. Comparison of Different CNN Methods

In order to demonstrate the superiority of our VGG16 network in image feature extraction and further emphasize the necessity of incorporating socio-economic data into the research, we conduct comparative experiments between VGG16 + BERT + RF (Random Forest) and other classical convolutional neural networks (CNNs). In the experiments, we directly train and test existing high-resolution remote sensing image scene classification methods, including AlexNet, ResNet, and DenseNet. The training and testing results obtained are shown in Figure 12.

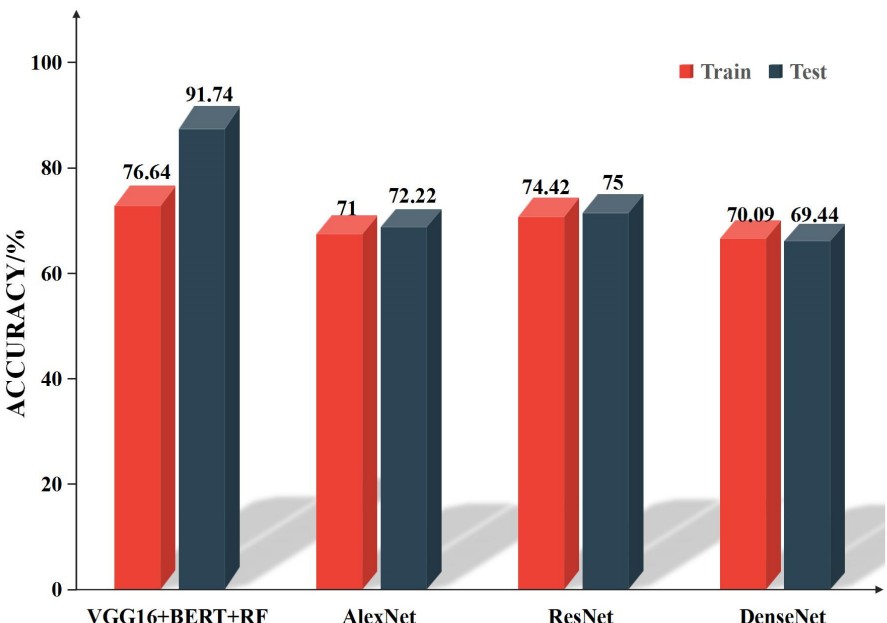

**Figure 12.** Training and testing results of different CNNs.

According to the results in Figure 12, the best training accuracy of VGG16 + BERT + RF is 76.64%, which is 5.64%, 2.22%, and 6.55% higher than AlexNet, ResNet, and DenseNet, respectively. The testing accuracy of VGG16 + BERT + RF is 91.74%, surpassing the other three networks. Among the testing results of the other three networks, ResNet performs the best with a classification accuracy of 75%, followed by AlexNet with a testing accuracy of 72.22%, and DenseNet exhibits the lowest performance with a testing accuracy of 69.44%. Generally, the testing accuracy after training is expected to be higher than the accuracy during the training process, but DenseNet is an exception to this norm, likely due to the influence of the network's structural characteristics.

In summary, the above results clearly indicate that VGG16 + BERT + RF excels in performance on the research area's image dataset. Additionally, the results of these three networks also demonstrate that relying solely on image features cannot fully achieve highly accurate recognition of urban functional regions.

### 4.4. Comparative Experiment with Traditional Recognition Methods

To emphasize the advantages of the novel urban functional region recognition method proposed in this paper, we conduct comparative experiments with traditional recognition methods and similar methods mentioned in reference [1]. Widely used traditional techniques such as K-Means clustering and semantic models based on LDA belong to unsupervised classification methods, while the latter falls under supervised classification. The five sets of comparative experiments include (1) single POI, (2) single building footprints, (3) combining POIs and building footprints, (4) combining POIs, building footprints, and remote sensing images, (5) combining POIs, building footprints, Weibo, and remote sensing images.

The results, as shown in Figure 13, show that (1) when using a single POI, K-means achieves an accuracy of 60.86%, LDA achieves 60.68%, and SOE achieves 66.73%; (2) when using only building footprints, K-means obtains 65.44%, LDA obtains 58.68%, and SOE obtains 57.58% accuracy; (3) combining features of POIs and building footprints, K-means achieves an accuracy of 69.92%, LDA achieves 66.37%, while SOE and ASOE achieve 69.67%; (4) combining POIs, building footprints, and imagery, K-means and LDA obtain 74.41% and 87.47%, while SOE obtains 90.36%; and (5) the accuracy rates of LDA, K-means and SOE methods based on ASOE features are 80.14%, 81.65%, and 90.36%, respectively, which are all lower than our method, with an accuracy rate of 91.74%. Traditional methods like LDA and K-means are not highly efficient at feature extraction for multi-source data fusion in classification tasks, and they struggle to effectively integrate different types of data, such as POIs, Weibo, and buildings. Meanwhile, SOE, due to its lack of consideration for incorporating Weibo semantic features, also yields lower classification accuracy compared to our approach. Therefore, through this comparison, we believe that our ASOE-based method is superior and achieves excellent classification results.

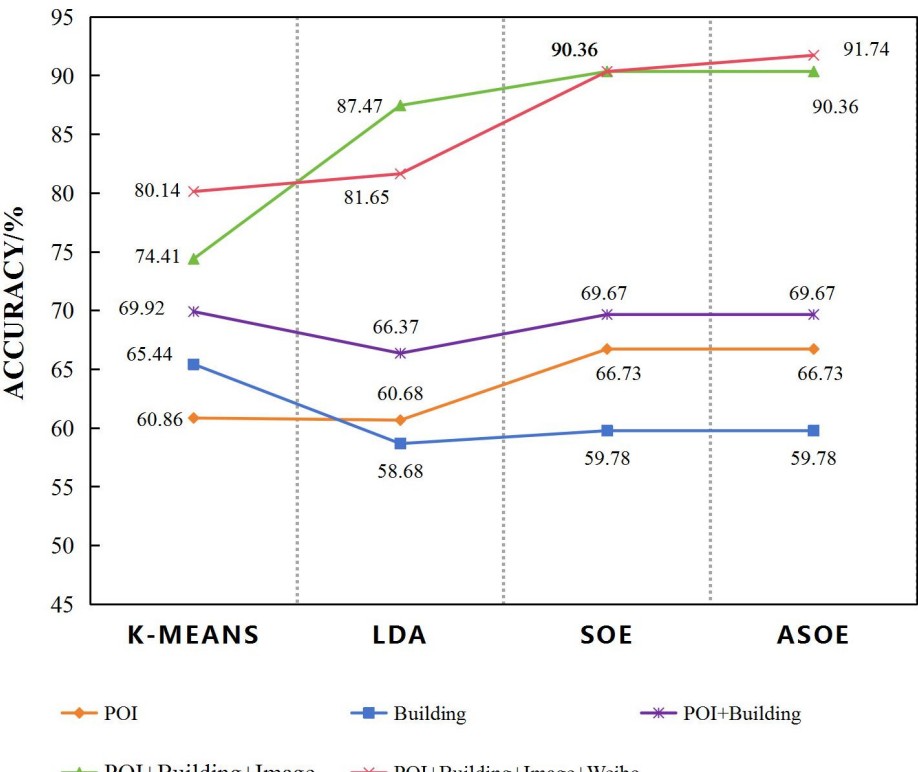

**Figure 13.** Comparison with traditional methods and SOE.

In the experiments, the Xi'an area within the EULUC-China map is selected as the research unit. In the first-level classification (residential, entertainment, transportation (not involved in validation), industrial, and high-rise), Gong Peng used samples representing over 70% of the dominant land use for training and validation [41], achieving an accuracy of 58.9%. In the research by Feng Ying [1] on identifying functional regions in Shenzhen, training and validation were conducted with over 60% of the samples, resulting in an accuracy of 90.94%. In our study, the accuracy reaches 91.74%. This indicates that our proposed integrated approach effectively extracts features of functional regions for classification.

## 5. Conclusions

Addressing the issues of insufficient data feature mining and the complexity of classification methods in traditional urban functional region identification, this paper first integrates the four types of big data mentioned above and proposes an ASOE learning framework. This method is applied to the main urban area of Xi'an, achieving an accuracy of 91.74% in identifying four typical functional regions. Furthermore, we employ a method from the Random Forest classifier to quantitatively calculate the weights of each factor in both single-data-source and multi-data-source fusion classification tasks, thereby highlighting the importance of each data feature. Additionally, we conduct comparative experiments between our method and traditional functional region identification approaches. The following conclusions can be drawn:

(1) The fusion method of multiple geospatial data sources leverages the advantages of big data, thoroughly extracting multidimensional data features that reflect functional region differences, thereby achieving higher accuracy in urban functional region classification and identification. What is more important, our approach can capture dynamic human activity characteristics and achieve a "from people to land" inversion process when compared to the SOE method that does not utilize social remote sensing data.

(2) In the final classification task, remote sensing images contribute the main spatial information. This is because the VGG16 network is capable of fully extracting hidden high-level semantic features, which play a decisive role in efficiently identifying functional regions.

Certainly, although this framework has achieved good results, there are still two aspects that require further in-depth research. Firstly, the framework is limited by the coverage range of Weibo data, which results in the inability to extract corresponding resident activity features for some land parcels. Similarly, mobile signaling data and taxi trajectory data, due to privacy concerns and other issues, are either inaccessible or incomplete. Secondly, this paper focuses on identifying typical individual functional regions and overlooks mixed-use regions. These mixed regions can be further subdivided, with samples extracted and input into the network for more detailed functional region delineation. Subsequent research can involve incorporating mobile signaling or trajectory data with wider coverage to identify urban functional regions on a larger scale and with greater precision.

**Author Contributions:** Conceptualization, Jianjun Bai; Data curation, Zhuo Wang; Formal analysis, Zhuo Wang; Funding acquisition, Jianjun Bai and Ruitao Feng; Investigation, Jianjun Bai; Methodology, Zhuo Wang; Project administration, Ruitao Feng; Software, Zhuo Wang; Supervision, Jianjun Bai and Ruitao Feng; Validation, Zhuo Wang and Ruitao Feng; Writing—original draft, Zhuo Wang; Writing—review and editing, Jianjun Bai and Ruitao Feng. All authors have read and agreed to the published version of the manuscript.

**Funding:** This research was funded by the National Natural Science Foundation of China under Grant 42271289; and the National Natural Science Foundation of China under Grant 42101341. And the APC was funded by Jianjun Bai and R.F.

**Data Availability Statement:** The data are not publicly available due to trade secrets and personal privacy restrictions.

**Acknowledgments:** This work was supported by the National Natural Science Foundation of China under Grant 42271289; and the National Natural Science Foundation of China under Grant 42101341. The authors would like to thank the organizations and individuals who provided publicly accessible data. The author would also like to thank the anonymous reviewers for their time and effort to review and improve this work.

**Conflicts of Interest:** The authors declare that they have no known competing financial interests or personal relationships that could have appeared to influence the work reported in this paper.

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
