# Peer review of "A Multi-Feature Fusion Method for Urban Functional Regions Identification: A Case Study of Xi’an, China"

_ijgi, doi:10.3390/ijgi13050156_

Round 1

Reviewer 1 Report

Comments and Suggestions for Authors

1.      The abstract needs to be rephrased and arranged to give readers a brief and focused idea about what will be accomplished in this research. The abstract usually consists of an introduction to the topic, the basic idea of the research, the methods or methodology of the research, the area of study, the results, and the most important conclusions.

 2.      Page 2 (lines: 50-51): The author stated that “However, due to the limitations of low image resolution and limited spectral features, it is still inadequate to achieve high-precision functional zone identification, as it struggles to distinguish different functional zones effectively.” at present satellite images have become available with high resolution, hence the above sentence must be reconsidered or rephrased.

 3.      The relevant previous research was not clearly referred to in the introduction, as the author referred very briefly to some previous research. This does not give readers a sufficient idea of what has been previously accomplished in this regard. The introduction also lacks a summary explaining the relationship of previous research to this research and how it can be used to achieve the main goal of this research.

 4.      Page 3 (lines 127-133): need reference.

 5.      Page 4 (lines 144-145): the author mentioned that “Parcels in EULUC-China [37] map are segmented using OSM road network, and we use them as basic units to identify functional zones.” The author used OpenStreetMap data, however, there are many studies indicating that there are some issues in the accuracy and quality of OSM data. There may be a loss or incomplete of some spatial information in it, which may affect the final results of the research. Justify, please?

 6.      Page 4 (lines 156-157): The author mentioned that the total number of POI data points was 344,990. However, this does not match the numbers in Figure 2, as the density in the legend reached 65,243,316!!!

 7.       Page 9 (lines 271-276): It is not clear in the paper why the original twenty categories of interest point data were reclassified into fourteen categories only??

8.     Page 14 (lines 369-379): Different percentages of classification accuracy were presented in Figure 10, however, the differences in these percentages have not been discussed in enough scientific manner. Explain in detail the main reasons that led to these differences.

9.      Pages 14 and 15 (Table 2 and Figure 11): same comments as the previous point.

Comments on the Quality of English Language

 Moderate editing of English language required

Reviewer 2 Report

Comments and Suggestions for Authors

This paper develops an urban functional area identification method named ASOE, leveraging a variety of multi-source data including remote sensing data, scraped building footprints, Points of Interest (POI), and Weibo data. This approach utilizes multi-source data to analyze urban areas. The article conducts solid and comprehensive discussions in its methods, results, and conclusion sections.

Recommendations for revisions before publication include:

1. Introduction meticulously reviews methods for urban functional area identification. However, the summary of existing research shortcomings at line 104 is somewhat abrupt. It is recommended to enhance persuasiveness by adding citations at this point.

2. Urban functional area identification involves diverse types of data. It is suggested to justify the use of POI, Weibo, and other data types in this study with relevant literature citations.

3. The method section provides extensive descriptions of data processing and methodological principles but lacks sufficient citations. It is advised to include more citations to reinforce the argument's credibility.

4. The style of literature citation within the article is inconsistent. A unified citation format is recommended.

5. Despite achieving a 91.74% identification accuracy, the paper acknowledges challenges such as limited generalizability, missing temporal data, and the inability to assess mixed-use areas. These issues represent current difficulties in urban functional area classification. The paper would gain additional value if it could make advancements in addressing these challenges.

Reviewer 3 Report

Comments and Suggestions for Authors

1.        L14, the VGG16 is the first to appear, it should be the full name.

2.        This accuracy is higher than other comparative methods.

3.        L142, it is not the main road network of Xi'an City, but the main road network of its main urban area.

4.       "Gaode Map" is not called Gaode, but AMap.

5.     The attributes of buildings, such as number of building floors, area and perimeter, etc., are difficult to obtain and were not clearly explained in the manuscript. Consider refer to the paper "High resolution mapping and evolution of steel stocks and waste in civil buildings: a case study of Changsha, China" to provide a more comprehensive overview of the field.

6.        L143-L165, OSM's road network, Google's Earth images, Amap's building footprints, and Baidu's POIs. How are these combined?

7.        L166, the main text does not lead to the kernel density map of POI. It is a bit awkward to appear directly here. What are the types of POI obtained? Those types should be listed in the table.

8.        Should be consistent of area, region and zone in all context.

9.        L171, the definitions of maybe is definition

10.     L179, points out that the model name is BERT, and generally the method is in uppercase letters. Other BERTs should also be modified.

11.     L188, Why use the VGG16 but not the VGG19

12.     Fig4 should have specific volume parameters. The Pooling in the legend may need to be changed to max Pooling?

13.     Fig7 length should be perimeter, and the middle pic does not have legends?

14.     Fig 8 is not a framework.

15.     Table5 should be three data sources.

16.     The accuracy unit % of Fig12 and Fig13 should be consistent, and the words in Fig12 are spelled incorrectly.

17.     The discussion is not in-depth enough.

18.    Revise conclusion to one or two paragraphs.

Comments on the Quality of English Language

The quality of English can be further improved.

Round 2

Reviewer 1 Report

Comments and Suggestions for Authors

Minor editing of English language required

Comments on the Quality of English Language

Minor editing of English language required

Reviewer 3 Report

Comments and Suggestions for Authors

1- The types and quantities of POI data does not match the POI classification of Baidu Maps in Table 1. The specific categories are not specified. The categories should be summarized according to Baidu's categories.

2- About Response 5, the author's specific modifications were not found, L177-L179 is Table1, and the author also did not make the required revision.

3- Cannot found the revision for Response 9.

4- Cannot found the revision for Response 15.

Comments on the Quality of English Language

Moderate editing of English language
